# Electric vehicle batteries alone could satisfy short-term grid storage demand by as early as 2030

Chengjian Xu [1] ✉, Paul Behrens [1], Paul Gasper [2], Kandler Smith [2], Mingming Hu[1], Arnold Tukker [1,3] & Bernhard Steubing [1]

The energy transition will require a rapid deployment of renewable energy (RE) and electric vehicles (EVs) where other transit modes are unavailable. EV batteries could complement RE generation by providing short-term grid services. However, estimating the market opportunity requires an understanding of many socio-technical parameters and constraints. We quantify the global EV battery capacity available for grid storage using an integrated model incorporating future EV battery deployment, battery degradation, and market participation. We include both in-use and end-of-vehicle-life use phases and find a technical capacity of 32–62 terawatt-hours by 2050. Low participation rates of 12%–43% are needed to provide short-term grid storage demand globally. Participation rates fall below 10% if half of EV batteries at end-of-vehicle-life are used as stationary storage. Short-term grid storage demand could be met as early as 2030 across most regions. Our estimates are generally conservative and offer a lower bound of future opportunities.

Electrification and the rapid deployment of renewable energy (RE) generation are both critical for a low-carbon energy transition[1,2]. They also address many other environmental issues, including air pollution. However, the variability of critical RE technologies, wind and solar, combined with increasing electrification may present a challenge to grid stability and security of supply[1,2]. There are several supply-side options for addressing these concerns: energy storage, firm electricity generators (such as nuclear or geothermal generators), long-distance electricity transmission, over-building of RE (resulting in curtailment in periods of lower demand), and power-to-gas[3] (in approximate ascending order of today's estimated cost). Demand-side management is also vital in shifting and flattening peak demand[4]. Given rapid cost-declines, battery storage is one of the major options for energy storage and can be used in various grid-related applications to improve grid performance. Cost declines in batteries have been the major driver for electric vehicle (EV) cost reductions. Given that many batteries will be produced for light-duty transport these could offer a low-cost and materially-efficient approach for short-term electricity grid storage requirements[5].

EV batteries can be used while in the vehicle via vehicle-to-grid approaches, or after the end of vehicle life (EoL) (when they are removed and used separately to the chassis in stationary storage). "Smart" vehicle-to-grid charging can facilitate dynamic EV charging and load shifting grid services. EVs can also be used to store electricity and deliver it back to the grid at peak times[6]. These opportunities rely on standards and market arrangements that allow for dynamic energy-pricing and the ability of owners to benefit from the value to the grid. Value to the grid can include deferred or avoided capital expenditure on additional stationary storage, power electronic infrastructure, transmission build-out, and more[6]. When the remaining battery capacity drops to between 70-80% of the original capacity, batteries generally become unsuitable for use in EVs[7]. However, these batteries at vehicle EoL (hereafter termed retired batteries) may still have years of useful life in less demanding stationary energy storage applications and represent substantial value to the grid[8].

The utilisation of EV batteries could improve the flexibility of supply while reducing the capital costs and material-related emissions associated with additional storage and power-electronic

[1]Institute of Environmental Sciences (CML), Leiden University, 2300 RA Leiden, The Netherlands. [2]National Renewable Energy Lab, 15013 Denver West Parkway, Golden, CO, USA. [3]Netherlands Organisation for Applied Scientific Research TNO, 2595 DA Den Haag, Netherlands. ✉e-mail: xuchegjian@gmail.com

infrastructure. However, the total grid storage capacity of EV batteries depends on different socioeconomic and technical factors such as business models, consumer behaviour (in driving and charging), battery degradation, and more[9,10]. Previous global-level studies, including those on vehicle-to-grid capacity[2,11,12] and retired battery capacity[12,13] are informative. However, they rarely consider several important factors that determine storage opportunity, such as non-linear, empirically-based battery degradation and neglect the impact of battery chemistry altogether;[14–16] geographical and/or temporal temperature variance (which impacts battery degradation); and, driving intensity by vehicle type in different countries/regions (which constrains the total capacity available during the day). Additionally, consumer participation in the vehicle-to-grid market and utilisation of retired batteries in the second-use market impact the actual grid storage capacity[10], both of which are important but rarely quantified.

Here we link three models and databases to assess the global grid storage opportunity of EV batteries by 2050 for both vehicle-to-grid applications and EoL opportunities (see Fig. 1, Methods, and Supplementary Fig. 1). We cover the main EV battery markets (China, India, EU, and US) explicitly, and combine other markets in a Rest of the World region (RoW). We first use a dynamic battery stock model to estimate future battery demand as part of transport fleets per region (Supplementary Fig. 2). The model incorporates two EV fleet development scenarios based on the IEA's (International Energy Agency), stated policy (STEP) and sustainable development (SD) scenarios. The STEP scenario incorporates existing EV policies only, while the SD scenario is compatible with the climate goals of the Paris agreement and sees a larger EV fleet. The scenarios include two battery chemistry sub-scenarios to capture different technological paths: one dominated

by lithium nickel cobalt oxides (NCX, with an "X" denoting manganese or aluminum, i.e., NMC/NCA) and another dominated by lithium-ion phosphate or (LFP). Market shares of NCX and LFP batteries are assumed to reach 98% and 2% in the NCX path by 2050, respectively, and 40% and 60% in the LFP path (see Supplementary Fig. 3 for detailed market shares over time).

These estimates of future demand are linked to an EV driving and charging behavior model for small, mid, and large-size BEVs (battery electric vehicles) and PHEVs (plug-in hybrid electric vehicles) based on daily driving distance distributions for different regions (Supplementary Figs. 4–6). EV use behavior, battery chemistry, and temperature in each region are combined with the latest battery degradation data for NCX[14,15,17] and LFP[16] chemistries to account for region- and chemistry-specific battery degradation (Supplementary Fig. 7).

We first analyze the technical capacity for short-term grid storage from vehicle-to-grid and second-use. We then analyze the impact of different factors on the real-world capacity. For example, we analyse in detail the impact of different rates of EV owner participation in vehicle-to-grid markets as well as the impact of different utilisation rates of retired EV batteries in stationary storage (see Fig. 1 and methods for further details). Finally, we compare the technical and real-world short-term storage capacities against scenarios for future storage requirements from the literature.

We focus here on short-term energy storage since this accounts for the majority of the required storage capacity[18] and EV batteries are not well suited for longer-term, seasonal storage due to self-discharging over time. Short-term energy storage demand is typically defined as a typical 4-hour storage system, referring to the ability of a storage system to operate at a capacity where the maximum power delivered from that storage over time can be maintained for 4 hours.

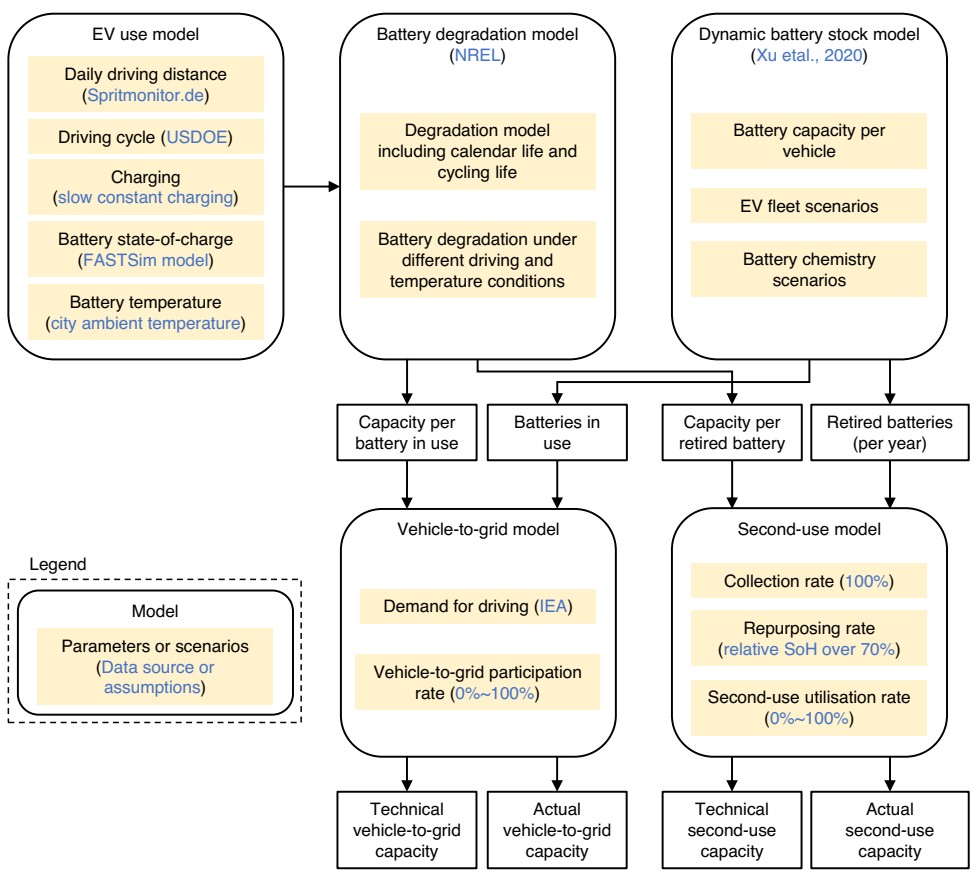

**Fig. 1 | Model framework linking EV use model, battery degradation model, and dynamic battery stock models.** See legend for use of colours. Square, white boxes indicate model outputs. Please see details for the model framework in the methods section. USDOE US Department of Energy, FASTSim Future Automotive Systems Technology Simulator, NREL National Renewable Energy Laboratory, IEA International Energy Agency, SoH State of Health.

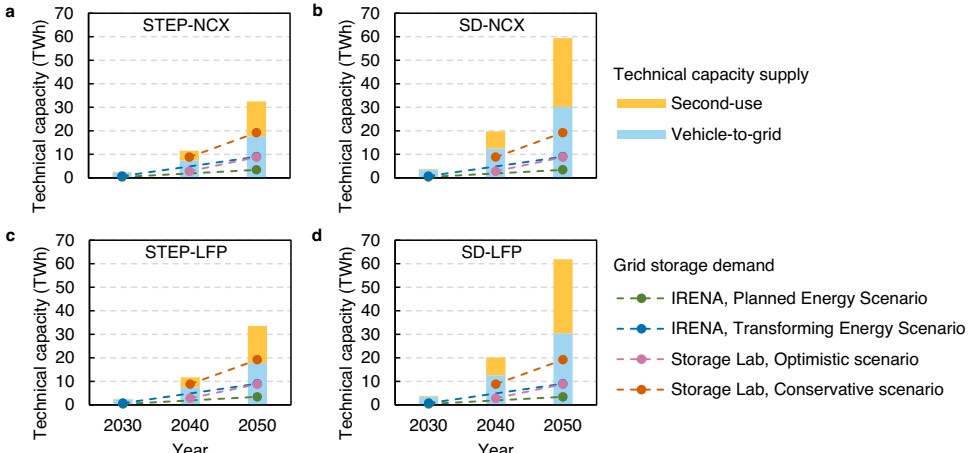

**Fig. 2 | Total technical capacity for EV batteries and comparison to grid storage demand. a** STEP-NCX scenario. **b** SD-NCX scenario. **c** STEP-LFP scenario. **d** SD-LFP scenario (see details in Supplementary Table 1). IRENA = International Renewable Energy Agency.

For example, the 4-hour storage capacity of batteries that together deliver a maximum of 0.25 GW until depletion will be 1 gigawatt hour[19] (GWh). The short-term storage capacity and power capacity are defined based on a typical 1-time equivalent full charging/discharge cycle per day (amounting to 4 hours of cumulative maximum discharge power per day). This 4-hour threshold is chosen as it is required by some jurisdictions such as the California Public Utilities Commission and New York Independent System Operator[20], energy system analysts anticipate this threshold as the most important to markets[21], and is often the length of time used in the literature[22].

We compare our results against storage requirements reported in the IRENA (International Renewable Energy Agency) Planned Energy and Transforming Energy Scenarios (with a warming of "likely 2.5 °C" and "well below 2 °C" in the second half of this century, respectively)[2], along with two Storage Lab scenarios (Conservative and Optimistic)[23]. Both Storage Lab scenarios result in a warming of "well below 2 °C" by 2100, but differ in the role for grid storage please see Supplementary Table 1 for more). These scenarios report short-term grid storage demands of 3.4, 9, 8.8, and 19.2 terawatt hours (TWh) for the IRENA Planned Energy, IRENA Transforming Energy, Storage Lab Conservative, and Storage Lab Optimistic scenarios, respectively. When assuming a 4-hour storage period for this capacity, this results in power demand of 850-4800 GW, or, 2500 GW when assuming an average storage capacity demand of 10 TWh.

## Results

### Total technical capacity

We define technical capacity as the total cumulative available EV battery capacity in use and in second use at a specific time while considering battery degradation and the capacity needed to meet driving demand. Globally, the SD scenario sees a total technical capacity twice that of the STEP scenario due to the larger fleet size (see Supplementary Fig. 8 and Note 1). Globally, the LFP scenario sees a slightly higher cumulative capacity than the NCX scenario, due to different battery market shares and the lower degradation of LFP across most countries/regions (see Supplementary Data 1 for a full comparison). Compared to the SD-NCX scenario, The SD-LFP scenario sees 2.6 TWh of higher technical capacity for China, EU, US, and RoW by 2050 compared to the SD-NCX and a 0.05 TWh lower technical capacity for India (see Supplementary Note 2). These capacity differences are small compared to the total technical capacity. As shown in Fig. 2, the SD-LFP scenario has a technical capacity 48% higher by 2030 and 91% higher than the STEP-NCX scenario by 2050 (3.8 TWh and 2.6 TWh in 2030 and 32 TWh and 62 TWh in 2050, respectively).

Under all scenarios, cumulative vehicle-to-grid and second-use capacity will grow dramatically, by a factor of 13–16 between 2030 and 2050. Putting this cumulative technical capacity into perspective against future demand for grid storage we find that our estimated growth is expected to increase as fast or even faster than short-term grid storage capacity demand in several projections[2,23] (Fig. 2). Technical vehicle-to-grid capacity or second-use capacity are each, on their own, sufficient to meet the short-term grid storage capacity demand of 3.4-19.2 TWh by 2050. This is also true on a regional basis where technical EV capacity meets regional grid storage capacity demand (see Supplementary Fig. 9).

### Vehicle-to-grid opportunities and limitations

Examining the vehicle-to-grid opportunity alone, we find that 21%-26% of the global theoretical battery capacity (i.e., on-board EV battery capacity of the entire EV fleet without considering battery degradation) could be available for vehicle-to-grid services by 2050 (Fig. 3a). The most important limiting factor is the battery capacity required to meet consumer driving demands[24,25]. Driving demand can limit the available capacity by 57%-63%. PHEVs, which make up around 11% of the theoretical capacity in 2050, are not considered for vehicle-to-grid as they have a low storage potential due to low capacities. On average, just 5% of the theoretical capacity is lost due to battery degradation by 2050. These losses vary between 7% in India and 4% in RoW due to differences in regional factors such as use conditions and temperature (for full regional results see Supplementary Fig. 10). Overall, taking these factors into account yields an estimated technical vehicle-to-grid capacity of 18-30 TWh by 2050 (see Fig. 3).

However, there are other factors that may limit real-world available storage capacity, primarily the vehicle-to-grid participation rate. Not all EV consumers will necessarily participate in the market and the participation rate is defined as the percentage of the technical vehicle to grid capacity connected to the grid, as shown in Fig. 3b. Participation rates of 38% and 20% are required to satisfy short-term storage demands of 10 TWh in 2050 (for STEP-NCX and SD-NCX scenarios, respectively). In practice, it is likely that EVs with high battery capacities and low degradation will be used for providing vehicle-to-grid services since these will provide the highest revenue for EV owners[26] (the full battery capacity distributions by 2050 across countries/regions are available in Supplementary Figs. 11–15).

### Impacts of deploying second-use batteries in stationary storage

Over time EV batteries degrade to the point they cannot be used to power vehicles[27], generally when the battery's relative State of Health

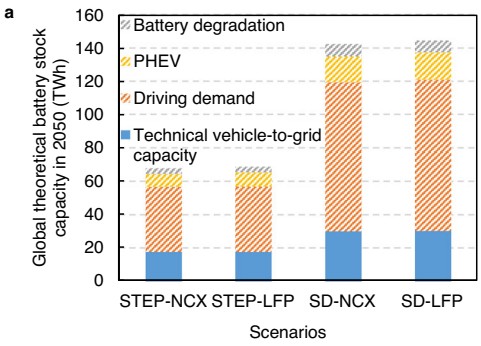
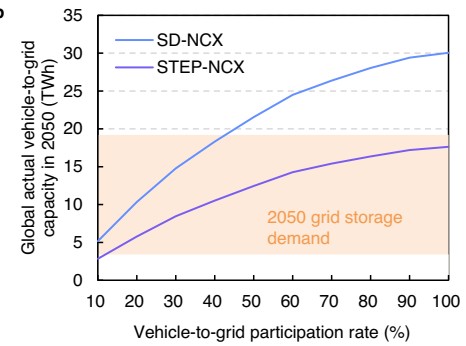

**Fig. 3 | Global available vehicle-to-grid capacity in 2050. a** Technical vehicle-to-grid capacity. Hatched bars indicate the capacity limits due to key factors and blue bars the technical vehicle-to-grid capacity. **b** Real-world vehicle-to-grid capacity as a function of participation rates. Results are shown for the STEP-NCX and the SD-NCX scenarios with a comparison to the range of storage demand computed by IRENA and Storage Lab models in 2050 (orange shading). Please see Supplementary Fig. 16 for global real-world vehicle-to-grid capacity under STEP-LFP and the SD-LFP scenarios and Supplementary Figs. 17–20 for regional real-world vehicle-to-grid capacity.

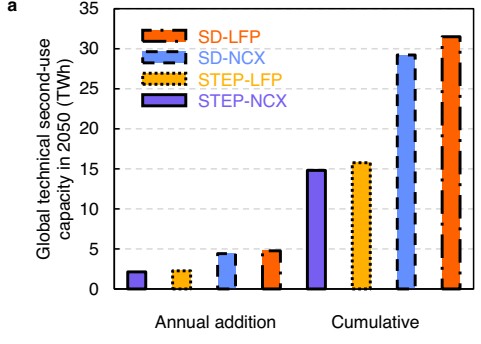
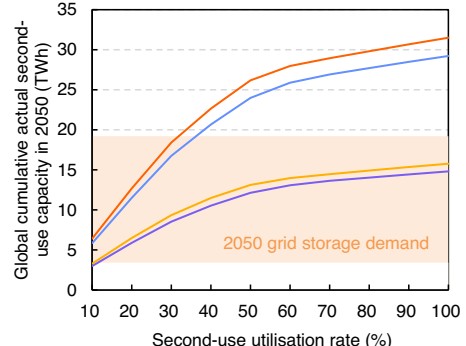

**Fig. 4 | Availability of second-use capacity globally in 2050. a** Average annual additions and cumulative technical capacity of second-use batteries in 2050. Here capacity refers to the technically available capacity considering battery degradation but without considering battery second-use utilisation rate. **b** Impacts of second-use utilisation rate on cumulative actual second-use capacity and a comparison to storage demand in 2050 (orange shading). See Supplementary Figs. 22–25 for regional actual second-use capacity.

(SoH) drops below 70%-80%[7] (defined as actual capacity as percentage of original capacity). The relative SoH could fall even lower if a consumer is willing to accept relatively poor battery health and shorter ranges[28]. Given their economic, value, size, and end-of-life regulations, we assume all batteries will be collected[29]. This is reasonable given that today's lead-acid batteries achieve a near 100% collection rate[30] and modern EV batteries are of much higher economic value.

Once collected, batteries are health tested to determine if they can be used in a less critical second-use application, or if they should be recycled[31]. Given the technical and economic feasibility of retired batteries for second-use[32], we consider batteries with an SoH of 70% and higher only for second-use (a threshold often used in the literature[32]). Under this assumption, 74% of retired NCX batteries can be repurposed for second-use globally, while 26% goes to recycling by 2050. Regional differences can be significant due to the impact of temperature on NCX battery degradation (see Supplementary Fig. 21 and Supplementary Data 1). In contrast, nearly all LFP retired batteries can be repurposed.

Business models are still developing, and repurposing is highly dependent on the technical specifications and market requirements of second-use applications[33]. Since battery disassembly is costly[32], battery repurposing will likely happen on the pack level instead of modules and cell level. Repurposing will consist mainly of rebalancing and reconnecting the retired battery packs. There is no strong technical reason to model a capacity difference before and after the repurposing.

Using these assumptions we find that 2.1–4.8 TWh of retired batteries are estimated to become available as annual technical second-use capacity globally in 2050, as shown in Fig. 4a. The cumulative technical second-use capacity is expected to reach 14.8–31.5 TWh by 2050 when assuming second-use batteries have a lifetime of 10-years[34] (Fig. 4b). The actual second second-use lifespan is uncertain due to uncertainties surrounding the retired battery SoH, use conditions, among other factors. Another uncertainty is the further battery degradation during secondary use, which is difficult to model due to complicated degradation mechanisms of retired batteries[35]. Further research into degradation and second-use life span is required to improve estimates of technical second-use capacity. If the 10TWh global, short-term storage requirements are met with second-use batteries alone, then a 68% utilisation rate of retired batteries would be needed in the STEP-NCX scenario (14.8 TWh technical capacity) and utilisation rate of 32% in the SD-LFP scenario (31.5 TWh technical capacity).

### Combining vehicle-to-grid participation rate and second-use utilisation rates

The global technical capacity for short-term grid storage of EV batteries grows rapidly in all scenarios. However, the real-world available capacity depends strongly on the vehicle-to-grid participation rate and the second-use utilisation rates. We show the real-world available capacity as a function of these rates in Fig. 5 (for the STEP-NCX scenario, please see Supplementary Figs. 26–28 for other scenarios).

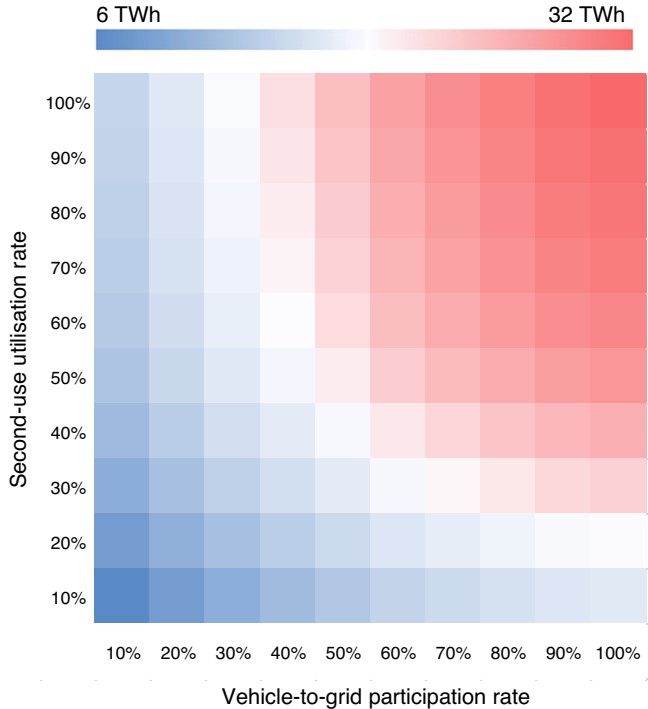

**Fig. 5 | Total actual available capacity under various conditions in STEP-NCX scenario in 2050.** Blue, white, and red colors depict minimum, average, and maximum values. See Supplementary Figs. 26–28 for other scenarios.

Participation and utilisation rates of 50% for vehicle-to-grid and second-use, results in a real-world capacity of 25–48 TWh by 2050, far higher than the short-term storage requirements estimated from the literature. Changes in vehicle-to-grid participation rates of 23–96%[36,37] by 2050 could influence this real-world capacity by as much as -24% to +21%. When second-use utilisation rates vary from 10%-100%, the real-world capacity varies between -41% and 12%. Taken together, vehicle-to-grid participation rate and second-use utilisation rate could alter the real-world capacity in 2050 by -61% to +32%.

We could see many different combinations of vehicle-to-grid and second-use to meet the short-term grid storage demands by 2050 (3.4–19.2 TWh). Without any second-use batteries in stationary storage, grids would require vehicle-to-grid participation rates of a modest 12–43%. If we assume that only half of second-use batteries are used on the grid (with others used off-grid, for other EV or storage purposes, etc.), the required participation rate of vehicle-to-grid drops to below 10%.

The required market participation rates depend on EV fleet and battery chemistry scenarios but also are influenced by other factors, such as battery capacity per vehicle. To investigate the impact of our capacity assumptions we investigate a scenario where all BEVs are equipped with a smaller 33kWh battery (instead of 33, 66, and 100 kWh battery per vehicle for small, mid-size, and large BEVs globally, see methods for more details). Even in this extreme case, EV batteries can still meet global, short-term grid storage demand by 2050 with participation rates of 10%-40% in vehicle-to-grid and with half second-use batteries used as stationary storage (see Supplementary Table 4).

## Discussion

Previous research has suggested that large EV fleets could exert additional stress on grid stability (e.g., if the majority of EVs are charged at grid peak time)[38]. Our findings reveal a different perspective that EV batteries could promote electricity grid stability via storage solutions from vehicle-to-grid and second-use applications. We estimate a total technical capacity of 32–62 TWh by 2050. This is

significantly higher than the 3.4–19.2 TWh required by 2050 in IRENA and Storage lab scenarios.

The real-world capacity depends on participation rates for vehicle-to-grid and utilisation rates for second-use of batteries. Participation rates may vary regionally depending on future market incentives and infrastructure, along with other factors[39]. The STEP-NCX scenario presented in Fig. 5 has the lowest technical capacity (32 TWh compared to 62 TWh in the SD-LFP scenario) which already easily meets requirements at participation rates of 40%-50% for vehicle-to-grid and with around half second-use batteries used as stationary storage. At a regional level, even lower participation rates may still contribute significantly to grid stability. Overall, EV batteries could meet short-term grid storage demand by as early as 2030 if we assume lower storage requirements from the literature and higher levels of participation and utilisation. By 2040–2050 storage demands are met across almost all scenarios and even low participation and utilisation rates.

Harnessing this potential will have critical implications for the energy transition and policymakers should be cognizant of the opportunities. The participation rate of EV users in the vehicle-to-grid market is crucial and the government can play an important role in incentivization. This can include market-based efforts such as micro-payments for services to the grid, or regulations to require the connection of commercial fleets to the network while at depots. Further regulations will be required to ensure the required hardware and software solutions for EV integration. This may include smart controllers for consumers in order to facilitate easy market participation and communication of benefits to EV users[39]. Strong re-use regulations will also be necessary to ensure that batteries are recovered at EOL and easily integrated into the grid[40]. Finally, policymakers and researchers should aim to understand EV user behavior over time in order to tackle the key factors preventing EV users from participating in vehicle-to-grid (which may include concerns surrounding battery degradation).

As we include a broader set of limitations for the total opportunity of EV storage our results are difficult to compare with other literature. Our estimated global EV fleet capacity in 2050 (68-144 TWh) is considerably higher than the estimate from IRENA (7.5-14 TWh)[2]. This is due to the IRENA's very conservative scenarios on future EV fleet size and battery capacity per vehicle. The IRENA scenario also does not consider the availability of EV fleet capacity for grid services. While a different IEA estimate does not extend beyond 2030[12] it does highlight the importance of including battery degradation in analyses, which we include for our projection to 2050 (Fig. 4).

We note several limitations in our approach that could be improved as data availability improves. For example, while we include battery degradation by using state-of-art data, future battery degradation is highly uncertain and depends on further technological breakthroughs both in battery chemistry such as Na-ion, Li-Air, and Li-Sulphur[41] along with developments in battery management systems. Further, while we derived driving behaviour from empirical data, future changes in driving habits are uncertain and dependent on various factors such as EV-related infrastructure. Vehicle chargers increase in power output over time and 50 kW charging and above is already common across many countries[42]. Frequent fast charging could lead to faster degradation, especially in hotter/colder climates[43]. This challenge may be addressed by future technology improvements to battery materials[44], electrode architectures, and optimized synergy of the cell/module/pack system design[45]. A further limitation is that we compare technical and real-world available vehicle-to-grid capacity with an average 4-hour storage requirement as provided in the scenarios by IRENA and Storage Lab. This omits potential differences in storage requirements at shorter time scales (seconds/minutes). Improved modelling and data can overcome this gap. It is however likely that the technical vehicle-to-grid capacity will be sufficient given

low vehicle utilisation rates of just 5% for many regions[46]. Additionally, the development of smart charging infrastructure and grid digitization is likely to provide additional flexibility for matching electricity demand and supply[47].

A final limitation is that we assume that the rated capacity per vehicle remains the same in the future and that a small number of large BEVs might provide large actual vehicle-to-grid capacity (Fig. 3). These capacities may change further in the future due to policy incentives, vehicle design, consumer preferences, charging infrastructure, among other factors. Further, the transportation system could see radical and fundamental changes. A significant and rapid shift away from private car use to mass transit, a move to shared electric vehicles, autonomous driving, and the success of battery swap systems[48] could all alter the available capacity by 2050.

In this study, we build a model framework to combine the EV use model, battery degradation model, and dynamic battery stock model. The model framework combines datasets on the real-world daily driving distance (in the EV use model), battery degradation test datasets (in the battery degradation model), and future EV and battery market data (in the dynamic battery stock model). The framework allows a structured use of diverse data to build a consistent perspective on future battery capacity. Within this model framework, this study provides a more complete understanding of the energy storage capacity available from EV batteries over time in real-world conditions and use. Results reveal a substantial opportunity for EV battery storage to support the stability and flexibility of renewable energy transition, even under modest consumer participation rates. To harness this opportunity, regulations and innovative business models will be needed to incentivize participation.

## Methods
### Model overview
We develop an integrated model to quantify the future EV battery capacity available for grid storage, including both vehicle-to-grid and second-use (see Supplementary Fig. 1 for an overall schematic). The integrated model includes three sub-models:

(1)   A dynamic battery stock model[27] to estimate total future EV battery stock and the retired batteries at vehicle EoL. This model considers EV fleet (i.e., battery stock) development and EV lifespan distribution (Supplementary Fig. 2), as well as future chemistry development (see Supplementary Fig. 3 for detailed battery market shares by chemistry).

(2)   An EV use model which includes behavioral factors such as EV driving cycle and charging behavior (changing power, time, and frequency), based on daily driving distance data for small/mid-size/large BEVs and PHEVs (Supplementary Figs. 4–6).

(3)   A battery degradation model based on the latest battery degradation test data, to estimate battery capacity fading over time under different EV use, battery chemistry, and temperature conditions (Supplementary Fig. 7).

### Dynamic battery stock model
We build on results and methods from the study[27] where we built a global dynamic battery stock model to quantify the stock and flows of EV batteries. We model future EV fleet development (i.e., battery stock) until 2050. We determine the retired battery availability based on battery stock development and EV lifespan distribution (which is assumed to determine the time when EV batteries are retired). Battery degradation does affect the technical performance (such as driving distance capability) of EVs, thus influencing consumers' choice of time when EVs come into EoL. Here, for model simplicity, we assume batteries will be retired only when EVs come into EoL. While for EV battery capacity, we use an average capacity of 33, 66, and 100 kWh for small/mid-size/large BEVs, and 21, 10, and 15 kWh for small/mid-size/large PHEVs.

We use two EV fleet scenarios until 2030 from the IEA: the stated policies (STEP) scenario and the sustainable development (SD) scenario. We further extend these two scenarios to 2050 based on a review of EV projections until 2050. We use the EV fleet share across 5 main EV markets (China, India, EU, US, and RoW) from the IEA until 2030, and keep the EV fleet share by countries/regions in 2030-2050 the same as the year 2030 due to lack of reliable data after 2030 (see Supplementary Data 1 for EV fleet scenarios by countries/regions). Further, we include 56 cities in China, 9 cities in India, 32 cities in EU, 53 cities in US, and 9 cities in RoW. We compile future EV sales share among 159 cities globally in STEP scenario and SD scenario based on future EV fleet projections by counties/regions from the IEA[25] and other data sources[49,50] (see Supplementary Data 1).

We consider battery market shares by chemistry based on the market share projections until 2030 from Avicenne Energy[51] a specialist consulting firm, and potential trends until 2050 from battery technology roadmaps[52–54] and commercial activities[55,56]. Current battery technology roadmaps issued by the US[52], EU[53], and China[54] focus on the development of high-energy Lithium Nickel Cobalt Manganese Oxide (transition to low cobalt and high nickel content) and Lithium Nickel Cobalt Aluminum-based chemistries. NCM and NCA batteries will likely make up the majority of next-generation EV Lithium-ion batteries. Future battery chemistry is uncertain after 2030. Existing Lithium Iron Phosphate batteries could also dominate the EV market, as indicated by recent commercial activities[55,56]. LFP battery manufacturers intend to improve the specific energy of LFP batteries to compete with NCM batteries[45]. Large-scale deployments of LFP may help avoid potential material supply shortage and price spikes associated with NCM and NCA batteries[27]. To encompass these market uncertainties, two battery chemistry scenarios are developed, including an NCX scenario (with X representing Manganese or Aluminum), and an LFP scenario. The market shares of NCX and LFP are assumed to reach 98% and 2% in the NCX path by 2050, and 40% and 60% in the LFP path (see Supplementary Fig. 3 for detailed battery market shares by chemistry in two scenarios).

### EV use model
We use the daily driving distance (DDD) of EVs based on data from Spritmonitor.de[24], an online quality-controlled, crowd-sourced database containing detailed real-world information on distances traveled, fuel consumption, and corresponding costs. It is widely used in the literature, including for estimation of the environmental impacts of vehicles[57] and the $CO_2$ mitigation potential of EVs[58]. We build historical DDD distributions for small/mid-size/large BEVs/PHEVs models, and explore the EV driving behavior of each EV model based on the corresponding DDD distributions. Please see the DDD distributions of each EV model in Supplementary Data 1. Note DDDs less than 5 km are excluded.

Further, we compile future DDD in different countries/regions (Supplementary Figs. 29–32) by assuming the future DDD is proportional to the future energy consumption per vehicle. The future energy consumption per vehicle in different countries/regions is estimated by the total EV fleet energy consumption divided by future EV fleet size in each country/region, which are both projected by the IEA[25].

By comparing various DDDs in multiples of EV range, we classify 5 DDD classes to formulate driving intensity and charging behavior. These 5 classes divided between 0% of the EV range to 200% of the EV range (i.e., a DDD twice the range of the EV) with intervals of 0–25%, 25–33%, 33–50%, 50–100%, 100–200%. We use the mean DDD of each class for calculations.

We assume two commuting trips between home and working place per day on weekdays and two entertaining trips on weekends for all countries/regions. Each trip distance is half of DDD. According to the required trip distance, we compile the driving cycle of each trip

(speed versus time) based on the standard US combined driving cycle (i.e., 55% city driving and 45% highway driving, see details in Supplementary Figs. 5 and 6, and Supplementary Note 1).

Charging behavior may be affected by charging infrastructure, amongst others, on-board EV charger, consumer preferences. We assume an immediate and slow home charging at constant charging power to full charge for all EV sizes and types because home charging is the major charging way (see Supplementary Data 1). We assume the home charging power as 1.92, 6.6, 22, and 1.92 kW for small, mid-size, large BEV, and PHEV, respectively[59]. We assume that due to high costs and limited utility no consumers will install higher power charging infrastructure at home. We further anticipate the charging behaviors in terms of changing frequency by comparing the various DDDs in multiples of the EV range. As driving intensity increases, the higher charging frequency is assumed for 5 DDD classes (1x every four days, 1x every three days, 1x every two days, 1x each day, and 2x every day respectively). For example, if the DDD of mid-size BEV (with a 312 km EV range) increases from 75 km to 625 km, and the battery needs to be charged more frequently from 1 time per four days to 2 times per day.

We calculate battery SoC under three EV states: driving, charging, and parked. For the battery SoC during driving, we use FASTSim model[59], Future Automotive Systems Technology Simulator developed by National Renewable Energy Laboratory (NREL), to calculate EV battery SoC second-by-second. The model inputs include the EV driving cycle, EV configurations, and battery performance parameters (specific energy and battery capacity). We select one representative EV model from the FASTSim model[59] for each EV size and type as EV configuration (Supplementary Table 2), and NCM622 as a representative chemistry for all EV types; because it was found that EV configurations and battery performance parameters (such as specific energy) had small effects on the resulting battery SoC simulations. For battery SoC during charging, we assume the battery SoC increases linearly under a constant charging power with a 90% charging efficiency[60]. If an EV is parked, the SoC of the battery is slowly decreasing due to losses caused by battery self-discharging. A typical self-discharging rate of 5% per month is assumed for lithium-ion battery[61]. Self-discharging occurs due to parasitic chemical reactions that consume active lithium and form electrochemically inactive species while lithium-ion batteries are at rest. These parasitic reactions both reduce the SoC of the cell, and also reduce the total amount of lithium available for cycling. The impact of self-discharge on the SoH of NCM and LFP batteries is captured in the battery degradation model we use. Note that for the sake of battery safety, a portion of battery capacity is unusable (15% for BEVs and 30% for PHEVs based on the BatPac model[62]), therefore we assume the usable SoC range as 5%-90% for BEV battery and 15%-85% for PHEV battery.

The battery temperature depends on the heat generation from chemical reactions inside batteries, amongst others, ambient temperature and environment (such as solar power radiation), battery management system (air or liquid cooling system to control battery temperature). The temperature can also vary from cell to cell, module to module, and component to component in the battery pack. The modelling of battery temperature is complicated and out of scope of this study. Here we use city ambient temperature to represent battery temperature, which is then used to calculate battery degradation. The main justification for this simplification is that the degradation for most consumer vehicles is dominated by calendar aging effects, as light-duty vehicles are only driven for a relatively small fraction of time throughout their life. For higher vehicle utilisation, neglecting battery pack thermal management in the degradation model will generally result in worse battery lifetimes, leading to a conservative estimate of electric vehicle lifetime. As such our modelling suggests a conservative lower bound of the potential for EV batteries to supply short-

term storage facilities. Here, we use monthly average temperature of total 159 cities to capture the effects of geographic and temporal temperature variance on battery degradation. The temperature data is collected from[63–66], can be found in Supplementary Data 1.

## Battery degradation model

Battery degradation is crucially important for determining EV battery capacity both in use and for second-life applications, but there are still many open research questions surrounding the importance of EV driving habits, charging behavior, and battery chemistries on capacity development[67]. Degradation model approaches include physics-based degradation models[68] as well as machine learning models[69,70] though there is no agreed-upon best practice[71]. Here, to balance the complexity and accuracy of the battery degradation model, we develop a semiempirical battery degradation model based on method from[16]. The model considers both calendar life and cycle life aging (Eq. (1)), assuming a square-root dependence on time for calendar life (degradation rates depend on temperature and SoC, see Eq. (2)) and a linear dependence on energy throughput for cycle life (degradation rates depend on temperature, Depth-of-Discharge (DoD), and Current rate ($C_{rate}$) see Eq. (3)).

$$q = 1 - q_{Loss,Calendar} - q_{Loss,Cycling} \tag{1}$$

$$q_{Loss,Calendar} = k_{Cal} \cdot \exp\left(\frac{-E_a}{\mathbf{R}T}\left(\frac{1}{T} - \frac{1}{\mathbf{T_{ref}}}\right)\right) \cdot \exp\left(\frac{\alpha \mathbf{F}}{\mathbf{R}}\left(\frac{U_a}{T} - \frac{U_{a,ref}}{\mathbf{T_{ref}}}\right)\right) \cdot \sqrt{t} \tag{2}$$

$$q_{Loss,Cycling} = k_{Cyc} \cdot (A \cdot DOD + B) \cdot (C \cdot C_{rate} + D) \cdot (G \cdot (T - \mathbf{T_{ref}})^2 + H) \cdot EFC \tag{3}$$

where q is the relative battery degradation, $q_{Loss,\,Calendar}$ is the relative calendar life degradation, $q_{Loss,\,Cycling}$ is the relative cycling life degradation, T is temperature, t is time (unit: days), EFC is equivalent full cycles. Note R is the universal gas constant (8.3144598 J mol⁻¹ K⁻¹), $T_{ref}$ is the reference temperature (298.15 K), F is Faraday constant (96485 C mol⁻¹), $k_{Cal}$ (unit: days⁰·⁵), $E_a$ (unit: J mol⁻¹ K⁻¹), and α (no unit) are fitting parameters for calendar life degradation, and $k_{Cyc}$ (unit: EFC⁻¹). A, B, C, D, G, and H (no units) are fitting parameters for cycling life degradation. The value of the anode-to-reference potential, $U_a$ (unit: V), is calculated from the storage SoC using the Eqs. (4) and (5)[72].

$$\begin{aligned} U_a(x_a) = {}& 0.6379 + 0.5416 \cdot \exp(-305.5309 \cdot x_a) + 0.044 \cdot \tanh\left(-\frac{x_a - 0.1958}{0.1088}\right) \\ & - 0.1978 \cdot \tanh\left(\frac{x_a - 1.0571}{0.0854}\right) - 0.6875 \cdot \tanh\left(\frac{x_a + 0.0117}{0.0529}\right) \\ & - 0.0175 \cdot \tanh\left(\frac{x_a - 0.5692}{0.0875}\right) \end{aligned} \tag{4}$$

where $x_a$, which represents the lithiation fraction of the graphite, is a simple linear function of the SoC:[73]

$$x_a(SOC) = x_{a,0} + SOC \cdot (x_{a,100} - x_{a,0}) \tag{5}$$

where $x_{a,\,0}$ is the lithiation fraction of the graphite at 0% SoC and $x_{a,\,100}$ is the lithiation fraction of the graphite at 100% SoC. $x_{a,\,0}$ equals to 0.0085, and $x_{a,\,100}$ equals to 0.78.

To obtain these fitting parameters, we collect publicly available battery degradation data, including calendar life aging and cycle life aging, for NCM[16] and LFP[14,15,17] chemistry. These data sets represent state-of-the-art lifetime performance for each chemistry; the LFP cells shown reach between 5000 and 8000 equivalent full cycles before reaching 80% remaining capacity, 4000-5000 equivalent full cycles

for NCM cells. This experimental data was then fit with the semi-empirical model Eqs. (1), (2), and (3) using a non-linear least squares solver in MATLAB. The NCM model has no $C_{rate}$ dependence, due to lack of data in the aging data set, so the parameters C and D are simply set at 0 and 1. We first fit the calendar fade data with the time-dependent portion of the model ($q_{Loss, Calendar}$, parameters $k_{Cal}$, $E_a$, and $\alpha$); the parameter $\alpha$ is bounded between -1 and 1, with other parameters unbounded. The parameters for the cycling fade (A, B, C, and D) are optimized on the cycling aging data. For both LFP and NCM, the raw cycling fade data was processed prior to optimizing a model based on expert judgement. For LFP, only cells with linear fade trajectories and data for at least 5000 EFCs were used for model optimization. For NCM, only data after 200 EFC at T > 5 °C and data at q < 0.85 at T < 5 °C was used for the optimization of the NCM cycling model parameters. The optimized parameters for the LFP and NCM degradation models are shown in Supplementary Table 3. Fitting results are shown in Supplementary Fig. 33 and degradation rates are shown in Supplementary Fig. 34.

Note that we assume NCA battery has the same degradation patterns as NCM battery due to a lack of state-of-the-art open-source data for NCA batteries. Besides cell chemistry, capacity degradation characteristics vary with cell design, manufacturing process, and proprietary additives[67,74], which is out of scope of this study. We use cell degradation patterns to represent battery pack degradation without consideration of cell-to-cell and module-module differences.

For simulation of the degradation under the EV driving loads (battery SoC evolution over time) and during dynamic temperature changes, the degradation model is reformulated to solve for the degradation occurring during consecutive timesteps[15]. We choose a timestep of 1 day for making SoH updates and update the SoC time-series for each day by the current SoH. At each timestep, the temperature is the average temperature during the simulation month at city from different countries/regions. Average SoC, DoD, $C_{rate}$, and the number of EFCs is extracted from the SoC timeseries. Average SoC refers to the time-averaged value of SoC. DoD is the difference between the maximum and minimum values of SoC. $C_{rate}$ is calculated using the absolute change of SoC per second, and then taking the average of all $C_{rates}$ greater than 0 during the entire timeseries. The number of EFCs is calculated by summing the changes to SoC over the timeseries. Dependence of the expected degradation rate on current SoH is incorporated by calculating a 'virtual time'[15]. The virtual time is found by inverting the calendar degradation equation to solve for time:

$$t_{virtual} = \left( q_{Current} / k_{Cal} \cdot \exp\left(\frac{-E_a}{\mathbf{R}T} \cdot \left(\frac{1}{T} - \frac{1}{\mathbf{T_{ref}}}\right)\right) \cdot \exp\left(\frac{\alpha \mathbf{F}}{\mathbf{R}} \cdot \left(\frac{U_a}{T} - \frac{U_{a,ref}}{\mathbf{T_{ref}}}\right)\right) \right)^2$$

(6)

The degradation change $\Delta q$ during any given timestep $\Delta t$ is then calculated by the following equation:

$$\Delta q = \left( k_{Cal} \cdot \exp\left(\frac{-E_a}{\mathbf{R}T} \cdot \left(\frac{1}{T} - \frac{1}{\mathbf{T_{ref}}}\right)\right) \cdot \exp\left(\frac{\alpha \mathbf{F}}{\mathbf{R}} \cdot \left(\frac{U_a}{T} - \frac{U_{a,ref}}{\mathbf{T_{ref}}}\right)\right) \right) / 2$$
$$\cdot \sqrt{t_{virtual} + \Delta t} \cdot \Delta t + k_{Cyc} \cdot (A \cdot DOD + B) \cdot (C \cdot C_{rate} + D)$$
$$\cdot (G \cdot (T - \mathbf{T_{ref}})^2 + H) \cdot \Delta EFC$$

(7)

For cycling fade, the virtual EFC does not need to be calculated, as the degradation rate is constant with respect to the change of EFC during any given timestep. This reformulation of the degradation model captures the path-dependent degradation observed in real-world battery use. See Supplementary Note 2 for modelled battery degradation for NCM and LFP.

## Available capacity from EV batteries

Vehicle EoL does not necessarily correspond to battery EoL. With technological improvements in battery reliability and durability, many batteries in EoL vehicles may still have years of useful life at the end of vehicle end of life. Vehicle battery EoL is usually as defined the time at which remaining battery capacity is between 70 and 80% of the original capacity[7]. We assume an EV lifespan distribution, used in our previous work[27], to account for EoL of EV. In our modelling approach, the vehicle lifespan distribution determines when batteries are not used in EVs any more (i.e., retired batteries). Retired batteries may have quite different capacity under different use conditions. When vehicles reach EoL due to consumer choices or other issues before the battery pack reaches 70% relative capacity, retired batteries will still have over 70% relative SoH and are assumed to be used in a second-life application. When battery pack reaches 70% relative SoH before a vehicle reaches its EoL, we assume that batteries may be still used in EVs for low distances-driving. Retired batteries from such vehicles will have lower than 70% relative SoH and are assumed to be recycled rather than for a second-use. We assume any battery with a relative SoH lower than 60% is recycled and removed from potential grid storage capacity[75]. However, even batteries with a relative SoH of 60–70% have a limited economic value and can have relatively high safety risks. (methods)[32].

We define technical vehicle-to-grid capacity as the availability of EV battery stock capacity for vehicle-to-grid application, considering the capacity reserved for EV driving, the capacity of PHEVs that will not participate in vehicle-to-grid due to low capacity, and capacity fade due to battery degradation. We further define the actual vehicle-to-grid capacity as the availability of technical vehicle-to-grid capacity for the grid under different consumer participation rates in the vehicle-to-grid business. Results focus on investigating under which participation rate can actual vehicle-to-grid capacity meet grid storage demand.

The technical second-use capacity is defined as the retired batteries capacity that can be repurposed (i.e., retired batteries with over 70% relative SoH). We further investigate actual second-use capacity under different utilisation rates (i.e., not all retired batteries will be deployed in second-use). The results are intended to determine the required utilisation rate for the second-use battery to meet grid storage demand.

We investigate the real-world capacity as a function of both vehicle-to-grid participation rate and second-use utilisation rates. We further analyze the market participation rates and utilisation rates that are required to meet short-term grid storage demand globally.

## Impact of battery capacity assumptions

The model is highly influenced by the battery capacity per vehicle. Therefore, we conduct a sensitivity analysis of battery capacity per vehicle by assuming all BEVs are small BEVs equipped with a battery with a capacity of 33 kWh. This assumption is based on three arguments: first, small BEVs could provide most of the daily driving demand for consumers, even though they have a lower driving range than large BEVs equipped with a high-capacity battery. Second, the development of widespread EV charging infrastructure, including fast charging technology, could help to overcome the range anxiety of small BEV owners. Third, the increasing use of small BEVs would reduce demand for batteries and materials, along with lowering embodied GHG emissions of those batteries.

## Data availability

The datasets, including EV fleet size by country, EV sales share by cities, and battery chemistry share, are all deposited in an Excel file (https://doi.org/10.6084/m9.figshare.21542472.v1). These raw data are used for the dynamic battery stock model for quantifying future battery flows. Please see the dynamic battery stock model from this link (https://doi.org/10.6084/m9.figshare.13042001.v4). City ambient temperature and

its effects on battery degradation are also deposited in the Excel file, while the code for estimating battery degradation, which is under privacy and license, is available upon reasonable request.

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

## Acknowledgements

C.X. acknowledges the China scholarship for financial support as well as Haixiang Lin and Gjalt Huppes for their useful recommendations to the study. P.G. and K.S. are supported by the National Renewable Energy Laboratory which is operated by Alliance for Sustainable Energy, LLC, for the U.S. Department of Energy under Contract No. DE-AC36-08GO28308, and acknowledge support from the Assistant Secretary for Energy Efficiency and Renewable Energy, Office of Vehicle Technologies of the U.S. Department of Energy. The views expressed in the article do not necessarily represent the views of the DOE or the U.S. Government. The U.S. Government retains and the publisher, by accepting the article for publication, acknowledges that the U.S. Government retains a nonexclusive, paid-up, irrevocable, worldwide license to publish or reproduce the published form of this work, or allow others to do so, for U.S. Government purposes.

## Author contributions

C.X. designed and conducted the research with valuable inputs from B.S., P.B., A.T., M. H., as well as P. G. and K. S. C.X. wrote the manuscript with the help of M.B., A.T., B.S., P. G., and other authors. P.B., A.T., and B.S. contribute significantly to the structure of research results and scientific writing of this research. P. G. and K. S. developed the battery degradation model, and further provided technical inputs on how to integrate the degradation model into the analysis of the results.

## Competing interests

The authors declare no competing interests.
