## [Peer Review File · Nature Communications]

Electric vehicle batteries alone could satisfy short-term grid storage demand by as early as 2030REVIEWER COMMENTS

Reviewer #1 (Remarks to the Author):

This paper presents a comprehensive research case highlighting the usage of EV battery storage can conveniently meet the global storage demand by 2030. Overall, it's an excellent read, and provides insight into many related and important considerations for determining the near-realistic total technical capacity of EV battery storage by 2030 and 2050. I have a few suggestions for improving the overall quality of the paper:

1. More discussion related to ENREF_50, ENREF_26,, and ENREF_60 must be added to the manuscript. Currently, it is confusing, and realizing its importance in determining the consumer driving demand and battery chemistry, I strongly suggest adding it.
2. Figures shown in the manuscript are observed to be exceeding the page margins. Please correct this. Also, in my humble opinion, for highlighting different scenarios, as shown in Fig. 3, different borders can be used to distinguish the cases, which becomes extremely difficult to comprehend in a black and white/grayscale print.
3. Please check the sentence on page 11, lines 350-351. I think the average capacity for small-size PHEVs is incorrect.
4. Please check the sentence on lines 416-417. I think self-charging is written instead of self-discharging.
5. Please add more discussion related to the typical self-discharging rate of 5% per month for LFP. A couple of sentences will be extremely helpful for readers to follow up.
6. Line 427-429: "Here we use city ambient temperature to represent battery temperature, which is then used to battery degradation." In my humble opinion, further details or references must be provided to justify this sentence, as the battery degradation is dictated heavily by the heat generation inside batteries and the management system/cooling (as mentioned). Suggesting the degradation of battery as a function of only ambient temperature negatively impacts the rest of the excellent work.
7. Lastly, adding a small paragraph highlighting the major contributions will be beneficial for the readers.

Reviewer #2 (Remarks to the Author):

This study estimates the market size of electric vehicle batteries to satisfy short term grid storage demand. The technical capacity for short term grid storage from vehicle to grid and second use is investigated. The authors have considered different factors like non-linear, empirically based battery degradation, geographical and/or temporal temperature variance, and driving intensity by vehicle type in different countries/regions. This is an interesting and useful study. Here are my comments:

- 1- Please check the writing of the paper again. For example, the sentence in line 124 of page 4 needs to be modified.
- 2- The authors need to add a figure to show the general framework of the study. The figure should contain: How they have collected data? what factors are considered for the EV? How the study has been done?
- 3- How long the electric vehicle batteries last for different countries by considering vehicle-to-grid?

Reviewer #3 (Remarks to the Author):

This paper quantifies the future EV battery capacity available for grid storage globally, considering battery deployment, battery degradation, and consumer participation rates in the vehicle-to-grid and second-use markets. I appreciate the authors' efforts in conducting this study, especially since the public is worried that vehicle electrification would stress grid stability. However, I think this paper can be strengthened further by improving the methodology, justifying research novelty and assumptions, and clarifying how their findings can advance our scientific understanding of this issue. Below are some specific suggestions:

- In the Introduction section, the authors should explain a few more sentences about scenarios (i.e., STEP and SD) definitions and the battery chemistry market share of the NCX and LFP technical paths. Otherwise, the readers are hard to follow.
- The author claimed that “We choose the industry standard, 4-hour storage capacity on a daily 92 basis, as EV batteries are unsuitable for longer-term, seasonal storage due to their 93 chemistries and use cases.” However, I could not find the relevant references regarding this statement.
- Figure 1 compares the estimated technical capacity between NCX and LFP. However, the authors did not explain what facts or assumptions cause the differences. Any implications from this comparison? It seems that (a) and (b) are similar to (c) and (d).
- The V2G or second-use participation rates are very critical to the available storage capacity results. Are there any studies exploring the relevant potential market? The authors estimate the minimum required participation rates under different scenarios, but in practice, how can the governments incentivize EV users to get involved? Readers would appreciate it if the author could discuss this in the paper.
- The authors indicate that 12%-43% market participation rates are required to meet short-term grid storage demand globally. However, the stated rates depend significantly on the EV capacity. What are the assumptions about the future EV capacity in the different regions? Sensitivity analysis may be helpful in this study, considering that the main results are obtained based on many assumptions.
- What are the major novelties in this study? The originality of the paper needs to be further clarified. It is of importance to have sufficient results to justify the novelty of a high quality journal paper.

Reviewer #1

Remarks	Responses
This paper presents a comprehensive research case highlighting the usage of EV battery storage can conveniently meet the global storage demand by 2030. Overall, it's an excellent read, and provides, insight into many related and important considerations for determining the near-realistic total technical capacity of EV battery storage by 2030 and 2050.	Thank you for your kind words about the manuscript.
More discussion related to ENREF_50, ENREF_26,, and ENREF_60 must be added to the manuscript. Currently, it is confusing, and realizing its importance in determining the consumer driving demand and battery chemistry, I strongly suggest adding it.	We have added further discussion (L421-425) describing the online Spritmonitor.de database and we have explained why this is an excellent approach for determining consumer driving demand: “We use the daily driving distance (DDD) of EVs based on data from Spritmonitor.de¹, an online quality-controlled, crowd-sourced database containing detailed real-world information on distances traveled, fuel consumption, and corresponding costs. It is widely used in the literature, including for estimation of the environmental impacts of vehicles² and the CO₂ mitigation potential of EVs³.” We have also added further discussion on NREF_26 and ENREF_60, (L402-419) with respect to future battery chemistry. The section now reads: “We consider battery market shares by chemistry based on the market share projections until 2030 from Avicenne Energy⁴ a specialist consulting firm, and potential trends until 2050 from battery technology roadmaps⁵⁻⁷ and commercial activities^{8,9}. Current battery technology roadmaps issued by the US⁵, EU⁶, and China⁷ focus on the development of high-energy Lithium Nickel Cobalt Manganese Oxide (transition to low cobalt and high nickel content) and Lithium Nickel Cobalt Aluminum-based chemistries. NCM and NCA batteries will likely make up the majority of next-generation EV Lithium-ion

	batteries. Future battery chemistry is uncertain after 2030. Existing Lithium Iron Phosphate batteries could also dominate the EV market, as indicated by recent commercial activities^{8,9}. LFP battery manufacturers intend to improve the specific energy of LFP batteries to compete with NCM batteries¹⁰. Large-scale deployments of LFP may help avoid potential material supply shortage and price spikes associated with NCM and NCA batteries¹¹. To encompass these market uncertainties, two battery chemistry scenarios are developed, including an NCX scenario (with X representing Manganese or Aluminum), and an LFP scenario. The market shares of NCX and LFP are assumed to reach 98% and 2% in the NCX path by 2050, and 40% and 60% in the LFP path (see Supplementary Fig. 3 for detailed battery market shares by chemistry in two scenarios).”
Figures shown in the manuscript are observed to be exceeding the page margins. Please correct this. Also, in my humble opinion, for highlighting different scenarios, as shown in Fig. 3, different borders can be used to distinguish the cases, which becomes extremely difficult to comprehend in a black and white/grayscale print.	We have edited the document to ensure that all figures lie within the page margin. Thank you for the suggestion on Fig. 3. We have added different borders to Fig 3.. We have also checked legibility by converting Fig. 3 to grayscale to check if the bars are distinguishable.
Please check the sentence on page 11, lines 350-351. I think the average capacity for small-size PHEVs is incorrect.	Thank you for checking this. We have made a further review of the literature and we think the estimate of 21 kWh is correct. For instance, the Cadillac ELR and Chevrolet Volt are both classified as PHEVs since their interior passenger and cargo volumes are less than 110 cubic feet, according to US EPA vehicle size classes. The Cadillac ELR is equipped with a 16.5 (2014)–17.1 (2016) kWh lithium-ion battery, and Chevrolet Volt with a 18.4 kWh (2016–2019) lithium-ion battery. These real numbers are a little bit lower than our estimation of 21 kWh. This is because, in our estimation, we incorporate different small-size PHEV models and make a market sales-based average value. According to previous work, EV capacity is estimated based on the technical specifications of EVs, which can vary a lot from one EV model to another. To reach our

	value we first estimate the small-size PHEV battery capacity of each model, which is calculated as the all-electric range (miles) * fuel economy (Wh/mile) / 1000 / 0.7 (0.7 is the availability of battery capacity for driving). Taking this into account and the market sales of each small-size PHEV model, we further estimate a market sales-based average capacity for small-size PHEV. This may result in a higher battery capacity for small PHEVs than for mid-size and large PHEVs, depending on the PHEV market dynamics and manufacturers' battery design.
Please check the sentence on lines 416-417. I think self-charging is written instead of self-discharging.	Thank you for highlighting this. We have double-checked and it says self-discharging (see L104 and L475-482) We also see that the statement of three EV states in L465-466 could lead to confusion. The end of that sentence has been changed to: "We calculate battery SoC under three EV states: driving, charging, and parked."
Please add more discussion related to the typical self-discharging rate of 5% per month for LFP. A couple of sentences will be extremely helpful for readers to follow up.	Thank you for this suggestion. We have added further discussion reading (L477-482): "Self-discharging occurs due to parasitic chemical reactions that consume active lithium and form electrochemically inactive species while lithium-ion batteries are at rest. These parasitic reactions both reduce the SoC of the cell, and also reduce the total amount of lithium available for cycling. The impact of self-discharge on the SoH of NCM and LFP batteries is captured in the battery degradation model we use."
Line 427-429: "Here we use city ambient temperature to represent battery temperature, which is then used to battery degradation." In my humble opinion, further details or references must be provided to justify this sentence, as the battery degradation is dictated heavily by the heat generation inside batteries and the management system/cooling (as mentioned). Suggesting the	Thank you. You are correct in noting that thermal management is an important factor in estimating battery lifetime. To address this we have added further text (L494-501) to the 'EV use model' subsection: "The main justification for this simplification is that the degradation for most consumer vehicles is dominated by calendar aging effects, as light-duty vehicles are only driven for a relatively small fraction of time throughout their life. For higher vehicle utilisation, neglecting battery pack thermal management in the degradation model will generally result in worse battery lifetimes, leading to a conservative estimate of electric vehicle lifetime. As such

degradation of battery as a function of only ambient temperature negatively impacts the rest of the excellent work.	our modelling suggests a conservative lower bound of the potential for EV batteries to supply short-term storage facilities.”
Lastly, adding a small paragraph highlighting the major contributions will be beneficial for the readers.	Thank you for the suggestion. We have added the following paragraph (L349-360): “In this study, we build a model framework to combine the EV use model, battery degradation model, and dynamic battery stock model. The model framework combines datasets on the real-world daily driving distance (in the EV use model), battery degradation test datasets (in the battery degradation model), and future EV and battery market data (in the dynamic battery stock model). The framework allows a structured use of diverse data to build a consistent perspective on future battery capacity. Within this model framework, this study provides a more complete understanding of the energy storage capacity available from EV batteries over time in real-world conditions and use. Results reveal a substantial opportunity for EV battery storage to support the stability and flexibility of renewable energy transition, even under modest consumer participation rates. To harness this opportunity, regulations and innovative business models will be needed to incentivize participation.”

Reviewer #2

Remarks	Responses
This study estimates the market size of electric vehicle batteries to satisfy short term grid storage demand. The technical capacity for short term grid storage from vehicle to grid and second use is investigated. The authors have considered different factors like non-linear, empirically based battery degradation, geographical and/or temporal temperature variance, and driving intensity by vehicle type in different countries/regions. This is an interesting and useful	Thank you for your kind remarks on the manuscript.

study.	
Please check the writing of the paper again. For example, the sentence in line 124 of page 4 needs to be modified.	Thank you for this suggestion. We have checked the paper and edited the text further. With regards to line 124 (now L122-127) we have updated it to read: “These scenarios report short-term grid storage demands of 3.4, 9, 8.8, and 19.2 terawatt-hour (TWh) for the IRENA Planned Energy, IRENA Transforming Energy, Storage Lab Conservative, and Storage Lab Optimistic scenarios, respectively. When assuming a 4-hour storage period for this capacity, this results in power demand of 850-4800 GW, or, 2500 GW when assuming an average storage capacity demand of 10 TWh.” We already checked and revised the writing of the sentence you pointed out (L122-127). The other major checks and revisions regarding the writing of this paper are described as follows: In L11-15, “We include both in-use and end-of-vehicle-life use phases and find a technical capacity of 32-62 terawatt-hours by 2050. Low participation rates of 12%-43% are needed to provide short-term grid storage demand globally. Participation rates fall below 10% if half of EV batteries at end-of-vehicle-life are used as stationary storage.” In L23-33, “There are several supply-side options for addressing these concerns including: energy storage, firm electricity generators (such as nuclear or geothermal generators), long-distance electricity transmission, overbuilding of RE (resulting in curtailment in periods of lower demand), and power-to-gas¹² (in approximate ascending order of today’s estimated cost). Demand-side management is also vital in shifting and flattening peak demand¹³. Given rapid cost-declines, battery storage is one of the major options for energy storage and can be used in various grid-related applications to improve grid performance. Cost declines in batteries has been the major driver for electric vehicles (EV) cost reductions. Given that many batteries will be produced for light-duty transport these could offer a low-cost and materially-efficient approach for short-term electricity grid storage requirements¹⁴.”

In L37-39, ““Smart” vehicle-to-grid charging can facilitate dynamic EV charging and load shifting grid services. EVs can also be used to store electricity and deliver it back to the grid at peak times¹⁵.”

In L43-47, “When the remaining battery capacity drops to between 70-80% of the original capacity, batteries generally become unsuitable for use in EVs¹⁶. However, these batteries at vehicle EoL (hereafter termed retired batteries) may still have years of useful life in less demanding stationary energy storage applications and represent substantial value to the grid¹⁷.”

In L102-115, “We focus here on short-term energy storage since this accounts for the majority of the required storage capacity¹⁸ and EV batteries are not well suited for longer-term, seasonal storage due to self-discharging over time. Short-term energy storage demand is typically defined as a typical 4-hour storage system, referring to the ability of a storage system to operate at a capacity where the maximum power delivered from that storage over time can be maintained for 4 hours. For example, the 4-hour storage capacity of batteries that together deliver a maximum of 0.25 GW until depletion will be 1-Gigawatt hour¹⁹ (GWh). The short-term storage capacity and power capacity are defined based on a typical 1-time equivalent full charging/discharge cycle per day (amounting to 4 hours of cumulative maximum discharge power per day). This 4-hour threshold is chosen as it is required by some jurisdictions such as the California Public Utilities Commission and New York Independent System Operator²⁰, energy system analysts anticipate this threshold as the most important to markets²¹, and is often the length of time used in the literature²².”

In L193-200, “**Impacts of deploying second-use batteries in stationary storage.** Over time EV batteries degrade to the point they cannot be used to power vehicles¹¹, generally when the battery relative State of Health (SoH) drops below 70%-80%¹⁶ (defined as actual capacity as percentage of original capacity). The relative SoH could fall even lower if a consumer is willing to

	accept relatively poor battery health and shorter ranges²³. Given their economic, value, size, and end-of-life regulations, we assume all batteries will be collected²⁴. This is reasonable given that today's lead-acid batteries achieve a near 100% collection rate²⁵ and modern EV batteries are of much higher economic value." In L254-259, "We could see many different combinations of vehicle-to-grid and second-use to meet the short-term grid storage demands by 2050 (3.4-19.2 TWh). Without any second-use batteries in stationary storage, grids would require vehicle-to-grid participation rates of a modest 12%-43%. If we assume that only half of second-use batteries are used on the grid (with others used off-grid, for other EV or storage purposes etc.), the required participation rate of vehicle-to-grid drops to below 10%."
The authors need to add a figure to show the general framework of the study. The figure should contain: How they have collected data? what factors are considered for the EV? How the study has been done?	Thank you for this suggestion. To make the framework clearer we have added a new figure (Fig. 1) in the main text. This figure includes the model types, data used, and the linking between the sub-models.
How long the electric vehicle batteries last for different countries by considering vehicle-to-grid?	We interpret your question as follows – if vehicle-to-grid use of batteries influences battery lifetimes. In this study, we assume that battery lifetime is not affected by vehicle-to-grid use, and do not further investigate this. Literature suggests vehicle-to-grid usage has small effects on battery lifetime in the first place²⁶⁻²⁸. Furthermore, an investigation would require taking into account that vehicle-to-grid use profiles vary from one application to another which is a study in itself. Within our modeled range of battery use conditions, including driving intensity and temperature, LFP batteries last on average 20 years until degradation to 70% of original capacity for all countries/regions. NCM batteries last on average 11, 20, 12, and 20 years until 70% degradation for China, India, EU, and US respectively.

Reviewer #3 (Remarks to the Author):

Remarks	Responses
This paper quantifies the future	Thank you for this remark and your suggestions for

EV battery capacity available for grid storage globally, considering battery deployment, battery degradation, and consumer participation rates in the vehicle-to-grid and second-use markets. I appreciate the authors' efforts in conducting this study, especially since the public is worried that vehicle electrification would stress grid stability. However, I think this paper can be strengthened further by improving the methodology, justifying research novelty and assumptions, and clarifying how their findings can advance our scientific understanding of this issue.	improvements. We have improved the description of the methodology (Fig. 1), and the discussion of research novelty (L349-360), reading: “In this study, we build a model framework to combine the EV use model, battery degradation model, and dynamic battery stock model. The model framework combines datasets on the real-world daily driving distance (in the EV use model), battery degradation test datasets (in the battery degradation model), and future EV and battery market data (in the dynamic battery stock model). The framework allows a structured use of diverse data to build a consistent perspective on future battery capacity. Within this model framework, this study provides a more complete understanding of the energy storage capacity available from EV batteries over time in real-world conditions and use. Results reveal a substantial opportunity for EV battery storage to support the stability and flexibility of renewable energy transition, even under modest consumer participation rates. To harness this opportunity, regulations and innovative business models will be needed to incentivize participation.”
In the Introduction section, the authors should explain a few more sentences about scenarios (i.e., STEP and SD) definitions and the battery chemistry market share of the NCX and LFP technical paths. Otherwise, the readers are hard to follow.	Thank you for the suggestion. We have added further descriptions of the STEP and SD scenarios as well as NCX and LFP technical paths (L70-80): “The model incorporates two EV fleet development scenarios based on the IEA’s (International Energy Agency), stated policy (STEP) and sustainable development (SD) scenarios. The STEP scenario incorporates existing EV policies only, while the SD scenario is compatible with the climate goals of the Paris agreement and sees a larger EV fleet. The scenarios include two battery chemistry sub-scenarios to capture different technological paths: one dominated by lithium nickel cobalt oxides (NCX, with an “X” denoting manganese or aluminum, i.e., NMC/NCA) and another dominated by lithium-ion phosphate or (LFP). Market shares of NCX and LFP batteries are assumed to reach 98% and 2% in the NCX path by 2050, respectively, and 40% and 60% in the LFP path (see Supplementary Fig. 3 for detailed market shares over time).”
The author claimed that “We	Thank you for highlighting the need for further

choose the industry standard, 4-hour storage capacity on a daily 92 basis, as EV batteries are unsuitable for longer-term, seasonal storage due to their 93 chemistries and use cases.” However, I could not find the relevant references regarding this statement.	discussion. We have added a further description of why we assume a 4-hour storage duration, reading (L102-115): “We focus here on short-term energy storage since this accounts for the majority of the required storage capacity¹⁸ and EV batteries are not well suited for longer-term, seasonal storage due to self-discharging over time. Short-term energy storage demand is typically defined as a typical 4-hour storage system, referring to the ability of a storage system to operate at a capacity where the maximum power delivered from that storage over time can be maintained for 4 hours. For example, the 4-hour storage capacity of batteries that together deliver a maximum of 0.25 GW until depletion will be 1-Gigawatt hour¹⁹ (GWh). The short-term storage capacity and power capacity are defined based on a typical 1-time equivalent full charging/discharge cycle per day (amounting to 4 hours of cumulative maximum discharge power per day). This 4-hour threshold is chosen as it is required by some jurisdictions such as the California Public Utilities Commission and New York Independent System Operator²⁰, energy system analysts anticipate this threshold as the most important to markets²¹, and is often the length of time used in the literature²².”
Figure 1 compares the estimated technical capacity between NCX and LFP. However, the authors did not explain what facts or assumptions cause the differences. Any implications from this comparison? It seems that (a) and (b) are similar to (c) and (d).	Thank you for the suggestion to further analyze these results. In the manuscript, we add more explanations reading (L133-140): “Globally, the LFP scenario sees a slightly higher cumulative capacity than the NCX scenario, due to different battery market shares and the lower degradation of LFP across most countries/regions (see Supplementary Data for a full comparison). Compared to the SD-NCX scenario, The SD-LFP scenario sees 2.6 TWh of higher technical capacity for China, EU, US, and RoW by 2050 compared to the SD-NCX and a 0.05 TWh lower technical capacity for India (see Supplementary Note 2). These capacity differences are small compared to the total technical capacity.”
The V2G or second-use participation rates are very critical to the available storage capacity results. Are there any	To the best of our knowledge, there is limited research on the potential market of storage capacity from EV batteries for both V2G and second-use participation rates. We have added a further discussion to outline

studies exploring the relevant potential market? The authors estimate the minimum required participation rates under different scenarios, but in practice, how can the governments incentivize EV users to get involved? Readers would appreciate it if the author could discuss this in the paper.	some of the challenges (L296-308). “Harnessing this potential will have critical implications for the energy transition and policymakers should be cognizant of the opportunities. The participation rate of EV users in the vehicle-to-grid market is crucial and the government can play an important role in incentivization. This can include market-based efforts such as micro-payments for services to the grid, or regulations to require the connection of commercial fleets to the network while at depots. Further regulations will be required to ensure the required hardware and software solutions for EV integration. This may include smart controllers for consumers in order to facilitate easy market participation and communication of benefits to EV users²⁹. Strong re-use regulations will also be necessary to ensure that batteries are recovered at EOL and easily integrated into the grid³⁰. Finally, policy makers and researchers should aim to understand EV user behavior over time in order to tackle the key factors preventing EV users from participating in vehicle-to-grid (which may include concerns surrounding battery degradation).”
The authors indicate that 12%-43% market participation rates are required to meet short-term grid storage demand globally. However, the stated rates depend significantly on the EV capacity. What are the assumptions about the future EV capacity in the different regions? Sensitivity analysis may be helpful in this study, considering that the main results are obtained based on many assumptions.	Thank you for these suggestions. We assume the future (rated) capacity of batteries used in small, mid-size, and large battery electric vehicles (BEVs) as 33, 66, and 100 kWh per vehicle globally. This is based on the dynamic battery stock model developed in previous work¹¹. We split the future global EV size by five countries/regions (China, India, EU, US, and RoW), based on IEA's projection of EV fleet share until 2030. Depending on EV fleet size, the EV battery capacity (in-use capacity) of 32-55, 6-11, 15-35, 8-23, and 7-20 TWh are estimated for China, India, EU, US, and RoW in 2050 (without considering battery degradation). Based on these EV battery capacity assumptions, in the first submission we showed required market participation rates of 12%-43% in 2050. However, this actually referred to vehicle-to-grid opportunities only. Now we include both the capacity of vehicle-to-grid and second-use to estimate the required market participation rates (L254-259). “We could see many different combinations of vehicle-to-

grid and second-use to meet the short-term grid storage demands by 2050 (3.4-19.2 TWh). Without any second-use batteries in stationary storage, grids would require vehicle-to-grid participation rates of a modest 12%-43%. If we assume that only half of second-use batteries are used on the grid (with others used off-grid, for other EV or storage purposes *etc.*), the required participation rate of vehicle-to-grid drops to below 10%.”

Thank you also for the suggestion of conducting a sensitivity analysis. We have investigated the impact of using a lower battery capacity per vehicle by assuming all BEVs are small BEVs equipped with a battery with a capacity of 33 kWh. We explain this approach in the methods, reading (L628-637):

“The impact of battery capacity assumptions. The model is highly influenced by the battery capacity per vehicle. Therefore, we conduct a sensitivity analysis of battery capacity per vehicle by assuming all BEVs are small BEVs equipped with a battery with a capacity of 33 kWh. This assumption is based on three arguments: first, small BEVs could provide most of the daily driving demand for consumers, even though they have a lower driving range than large BEVs equipped with a high-capacity battery. Second, the development of widespread EV charging infrastructure, including fast charging technology, could help to overcome the range anxiety of small BEV owners. Third, the increasing use of small BEVs would reduce demand for batteries and materials, along with lowering embodied GHG emissions of those batteries.”

We present the findings of this sensitivity analysis in the manuscript reading (L261-269):

“The impact of battery capacity per vehicle. The required market participation rates depend on EV fleet and battery chemistry scenarios but also are influenced by other factors such as battery capacity per vehicle. To investigate the impact of our capacity assumptions we investigate a scenario where all BEVs are equipped with a smaller 33kWh battery (instead of 33, 66, and 100 kWh battery per vehicle for small, mid-size, and large BEVs

	globally, see methods for more details). Even in this extreme case, EV batteries can still meet global, short-term grid storage demand by 2050 with participation rates of 10%-40% in vehicle-to-grid and with half second-use batteries used as stationary storage (see Supplementary Table 4).”
What are the major novelties in this study? The originality of the paper needs to be further clarified. It is of importance to have sufficient results to justify the novelty of a high quality journal paper.	Thank you for the suggestion to further describe the novelty of the paper. We have added further discussion reading (L349-360): “In this study, we build a model framework to combine the EV use model, battery degradation model, and dynamic battery stock model. The model framework combines datasets on the real-world daily driving distance (in the EV use model), battery degradation test datasets (in the battery degradation model), and future EV and battery market data (in the dynamic battery stock model). The framework allows a structured use of diverse data to build a consistent perspective on future battery capacity. Within this model framework, this study provides a more complete understanding of the energy storage capacity available from EV batteries over time in real-world conditions and use. Results reveal a substantial opportunity for EV battery storage to support the stability and flexibility of renewable energy transition, even under modest consumer participation rates. To harness this opportunity, regulations and innovative business models will be needed to incentivize participation.”

References

- 1 *MPG and Cost Calculator and Tracker* (Spritmonitor, 2020).
<https://www.spritmonitor.de/en/>
- 2 Plötz, P., Funke, S. Á. & Jochem, P. Empirical Fuel Consumption and CO2 Emissions of Plug-In Hybrid Electric Vehicles. *Journal of Industrial Ecology* **22**, 773-784 (2018).
- 3 Plötz, P., Funke, S. A., Jochem, P. & Wietschel, M. CO2 Mitigation Potential of Plug-in Hybrid Electric Vehicles larger than expected. *Scientific Reports* **7**, 16493 (2017).
- 4 *The Rechargeable Battery Market and Main Trends 2018-2030* (Avicenne Energy, 2019).
<https://www.bpifrance.fr/content/download/76854/831358/file/02%20-%20Presentatio n%20Avicenne%20-%20Christophe%20Pillot%20-%2028%20Mai%202019.pdf>
- 5 *Electrochemical energy storage technical team roadmap* (USDRIVE, 2017).
<https://www.energy.gov/sites/prod/files/2017/11/f39/EESTT%20roadmap%202017-10-16%20Final.pdf>

- 6 *Inventing the sustainable batteries of the future* (BATTERY 2030+, 2020).
https://battery2030.eu/digitalAssets/816/c_816048-l_1-k_roadmap-27-march.pdf
- 7 Chen, K., Zhao, F., Hao, H. & Liu, Z. Selection of lithium-ion battery technologies for electric vehicles under China's new energy vehicle credit regulation. *Energy Procedia* **158**, 3038-3044 (2019).
- 8 *Tesla wins China approval to build Model 3 vehicles with LFP batteries: ministry - Reuters* (Reuters, 2020).
- 9 *LFP chemistry is emerging as the future of batteries* (Clean Future, 2020).
- 10 Yang, X.-G., Liu, T. & Wang, C.-Y. Thermally modulated lithium iron phosphate batteries for mass-market electric vehicles. *Nature Energy* **6**, 176-185 (2021).
- 11 Xu, C. et al. Future material demand for automotive lithium-based batteries. *Commun. Mater.* **1**, 99 (2020).
- 12 Lund, P. D., Lindgren, J., Mikkola, J. & Salpakari, J. Review of energy system flexibility measures to enable high levels of variable renewable electricity. *Renewable and Sustainable Energy Reviews* **45**, 785-807 (2015).
- 13 Palensky, P. & Dietrich, D. Demand Side Management: Demand Response, Intelligent Energy Systems, and Smart Loads. *IEEE Transactions on Industrial Informatics* **7**, 381-388 (2011).
- 14 Brown, T., Schlachtberger, D., Kies, A., Schramm, S. & Greiner, M. Synergies of sector coupling and transmission reinforcement in a cost-optimised, highly renewable European energy system. *Energy* **160**, 720-739 (2018).
- 15 Guille, C. & Gross, G. A conceptual framework for the vehicle-to-grid (V2G) implementation. *Energy Policy* **37**, 4379-4390 (2009).
- 16 *Identifying and Overcoming Critical Barriers to Widespread Second Use of PEV Batteries* (National Renewable Energy Lab, 2015). <https://www.osti.gov/biblio/1171780>
- 17 Haram, M. H. S. M. et al. Feasibility of utilising second life EV batteries: Applications, lifespan, economics, environmental impact, assessment, and challenges. *Alexandria Engineering Journal* **60**, 4517-4536 (2021).
- 18 Guerra, O. J. Beyond short-duration energy storage. *Nature Energy* **6**, 460-461 (2021).
- 19 *Energy Storage Grand Challenge: Energy Storage Market Report* (U.S. Department of Energy, 2020).
https://www.energy.gov/sites/prod/files/2020/12/f81/Energy%20Storage%20Market%20Report%202020_0.pdf
- 20 *The Potential for Battery Energy Storage to Provide Peaking Capacity in the United States* (NREL, 2019). <https://www.nrel.gov/docs/fy19osti/74184.pdf>
- 21 *Storage Futures Study: Economic Potential of Diurnal Storage in the U.S. Power Sector* (National Renewable Energy Laboratory, 2021).
<https://www.nrel.gov/docs/fy21osti/77449.pdf>
- 22 *Energy Storage Technology and Cost Characterization Report* (2019).
<https://www.osti.gov/servlets/purl/1573487>
- 23 Saxena, S., Le Floch, C., MacDonald, J. & Moura, S. Quantifying EV battery end-of-life through analysis of travel needs with vehicle powertrain models. *Journal of Power Sources* **282**, 265-276 (2015).
- 24 Harper, G. et al. Recycling lithium-ion batteries from electric vehicles. *Nature* **575**, 75-

- 86 (2019).
- 25 Gaines, L. The future of automotive lithium-ion battery recycling: Charting a sustainable course. *Sustainable Materials and Technologies* **1-2**, 2-7 (2014).
- 26 Petit, M., Prada, E. & Sauvant-Moynot, V. Development of an empirical aging model for Li-ion batteries and application to assess the impact of Vehicle-to-Grid strategies on battery lifetime. *Applied Energy* **172**, 398-407 (2016).
- 27 Wang, D., Coignard, J., Zeng, T., Zhang, C. & Saxena, S. Quantifying electric vehicle battery degradation from driving vs. vehicle-to-grid services. *Journal of Power Sources* **332**, 193-203 (2016).
- 28 Peterson, S. B., Apt, J. & Whitacre, J. F. Lithium-ion battery cell degradation resulting from realistic vehicle and vehicle-to-grid utilization. *Journal of Power Sources* **195**, 2385-2392 (2010).
- 29 *A Vision for a Sustainable Battery Value Chain in 2030* (Global Battery Alliance, 2019).
https://www.globalbattery.org/media/publications/WEF_A_Vision_for_a_Sustainable_Battery_Value_Chain_in_2030_Report.pdf
- 30 Bai, Y. et al. Energy and environmental aspects in recycling lithium-ion batteries: Concept of Battery Identity Global Passport. *Materials Today* **41**, 304-315 (2020).

REVIEWERS' COMMENTS

Reviewer #1 (Remarks to the Author):

The authors have successfully addressed the reviewer's comments. The reviewer does not have any more comments.

Reviewer #2 (Remarks to the Author):

Thanks for preparing the responses. I have no further comment.

Reviewer #3 (Remarks to the Author):

The authors have addressed all the comments and answered the technical questions I have for this paper. The paper has been significantly improved after revision. The revised version of the manuscript appears to be good and ready for publication as far as I can tell.